

# Development and validation of a risk prediction model for hypoproteinemia after adult cardiac valve surgery: implications for clinical care

Fang Wang, Zhen-zhen Su, Xiao-qian Guo, Man Li, Rui Wang and Yan-jun Xu

Department of Cardiac Surgery, The First Affiliated Hospital of USTC, Division of Life Science and Medicine, University of Science and Technology of China, Hefei, Anhui Province, China

Corresponding author
Fang Wang,
wf18963782603@sina.com

## ABSTRACT

**Objective**. To construct and validate a risk prediction model for hypoproteinemia in adults following cardiac valve surgery with cardiopulmonary bypass (CPB), providing medical staff with an effective tool for early identification and intervention.

**Methods**. This retrospective cohort study analyzed clinical data from 259 patients who underwent CPB-assisted heart valve surgery at the Department of Cardiovascular Surgery, First Affiliated Hospital of China University of Science and Technology, between January and December 2023. Patients were divided into two groups based on whether their serum albumin levels fell below 35 g/L within 48 hours postoperatively: the hypoproteinemia group ($n = 144$) and the non-hypoproteinemia group ($n = 115$). Least absolute shrinkage and selection operator (LASSO) regression was used to identify candidate predictors, followed by multivariate logistic regression to determine independent risk factors.

**Results**. Among the 259 patients, 144 developed hypoproteinemia, yielding an incidence rate of 55.60%. LASSO regression identified nine variables associated with hypoproteinemia, and multivariate logistic regression confirmed eight independent predictors. Hypertension, chest infection, frailty, and preoperative heart failure were identified as independent risk factors (OR > 1, $P < 0.05$), while higher body mass index (BMI), red blood cell (RBC) count at admission, platelet count at admission, and albumin level at admission were protective factors (OR < 1, $P < 0.05$). The predictive model constructed using the nine LASSO-selected variables demonstrated good discrimination, with an area under the ROC curve (AUC) of 0.823 (95% CI [0.774–0.873]). The Hosmer–Lemeshow test showed no significant difference between predicted and observed outcomes ($P = 0.737$), indicating good model calibration.

**Conclusion**. The incidence of postoperative hypoproteinemia in this cohort was 55.60%. The developed nomogram model, based on key clinical predictors, demonstrated strong calibration and discrimination, offering a practical tool for identifying patients at high risk of hypoproteinemia following valve surgery.

## INTRODUCTION

Cardiac valvular disease is a prevalent condition among adult heart diseases and is frequently encountered in cardiac surgical procedures. The extent of the lesion can affect either a single valve or multiple valves simultaneously (*Hollenberg, 2017*; *Clavel, Iung & Pibarot, 2017*). In Western developed countries, age-related degenerative valvular heart diseases, such as aortic stenosis and mitral regurgitation, are predominant. The reported incidence rates for these conditions are between 3% and 4% (*Clavel, Iung & Pibarot, 2017*; *Iung & Vahanian, 2014*). In developing countries, rheumatic heart valve disease is the predominant condition, with an incidence rate that ranges from 2% to 3% (*Iung & Vahanian, 2014*; *Runlin, 2018*; *China Cardiovascular Health Disease Report Compilation Group, 2020*). Conservative medical treatment can provide temporary relief for heart failure symptoms, such as fatigue and shortness of breath. However, it is not capable of halting the progression or degeneration of valvular disease, nor does it enhance the long-term outcomes for patients (*Xiaokang et al., 2022*). Compared to drug-based conservative treatment, cardiopulmonary bypass (CPB) valve replacement offers a direct approach to repairing or replacing diseased valves, which can effectively enhance heart function, particularly for patients with a suboptimal response to medication. However, postoperative complications associated with CPB valve replacement are not uncommon, and these can prolong the recovery period and impose an additional economic burden on patients (*Wenjun, 2021*). Hypoproteinemia is a prevalent complication following cardiac surgery that involves CPB, and it is strongly associated with various factors during the perioperative period. Currently, there is a relative scarcity of research on hypoproteinemia post-CPB heart valve surgery, and the underlying pathogenesis remains largely undefined (*Wenjun et al., 2022*). It is highly important for clinical medical staff to accurately identify high-risk patients for postoperative hypoproteinemia, to refine treatment plans, and to implement preemptive nutritional support measures. This approach can reduce the wasteful use of medical resources and contribute to the development of a risk prediction model for hypoproteinemia following valve replacement with cardiopulmonary bypass. Currently, there are no published studies on such a risk prediction model for hypoproteinemia post-CPB valve replacement, either domestically or internationally. In light of this gap, this study reviewed the clinical data of 259 patients who underwent CPB heart valve surgery in our research unit from January to December 2023. We identified the influencing factors for postoperative hypoproteinemia and developed a personalized risk prediction model. The goal is to provide a scientific foundation for clinical decision-making. The development of this risk prediction model is not only a theoretical advancement but also a practical tool for clinicians. By accurately predicting the risk of postoperative hypoproteinemia, the model can guide clinical interventions and influence treatment planning. For instance, patients identified as high-risk can receive preemptive nutritional support and closer monitoring, thereby reducing the incidence and severity of hypoproteinemia and improving overall postoperative outcomes.

## SUBJECTS AND METHODS

### Research objects

This study is a retrospective cohort study. Between January and December 2023, a total of 259 patients underwent heart valve surgery under general anesthesia, hypothermia, and CPB in the Department of Cardiovascular Surgery at the First Affiliated Hospital of China University of Science and Technology.

Inclusion criteria were set as follows: patients aged 18 years or older, with normal preoperative liver function, no history of long-term significant drug use, and those undergoing aortic valve and/or mitral valve replacement or repair, potentially combined with tricuspid valve repair.

Exclusion criteria included patients under 18 years of age, those with active cirrhosis, acute or chronic hepatitis, autoimmune diseases, or malignant tumors, as well as individuals with abnormal preoperative liver function or incomplete clinical data. The present study obtained approval from the Medical Ethics Committee of the First Affiliated Hospital of the University of Science and Technology of China (Approval number: 2024-RE-210). Informed consent from the study participants was waived by the Medical Ethics Committee at the First Affiliated Hospital of the University of Science and Technology of China, as the study was conducted retrospectively.

### Data collection methods

Patient data were systematically collected through our hospital's electronic medical record system and hospital information system (HIS), encompassing the following categories: (1) Demographic, comorbidity and clinical data: gender, age, education level, occupational status, body mass index (BMI), NRS 2002 nutritional score at admission, and the presence of comorbidities such as hypertension, diabetes, coronary heart disease, and cerebral infarction, as well as clinical indicators including chest infection, frailty, pre-operative heart failure, and hemodilution on CPB. Specifically, frailty evaluation was conducted using an HIS-embedded electronic questionnaire, based on the FRAIL scale recommended by the International Academy of Nutrition and Aging (IANA) and adapted for the Chinese population. The scale encompasses five domains: physical fatigue, decreased resistance, reduced mobility, increased susceptibility to illness, and unintentional weight loss. Each domain is scored one point, with a total score of 3 or higher indicating frailty (*Dong et al., 2018*).

(2) Preoperative laboratory values: hemoglobin, albumin, white blood cell count, red blood cell count, platelet count, as well as levels of blood potassium, sodium, calcium, creatinine, and uric acid at admission;

(3) Preoperative echocardiographic measurements: left ventricular ejection fraction (LVEF), left atrial anteroposterior diameter, and left ventricular end-systolic diameter.

In accordance with the classification criteria from previous research (*Huimei et al., 2023*), subjects were categorized into two groups based on serum albumin concentration below 35 g/L within 48 h post-operation: the non-hypoproteinemia group with 115 cases and the hypoproteinemia group with 144 cases.

## Statistical methods

Data analysis was conducted using statistical software SPSS version 21.0 (IBM Corp., Armonk, NY, USA) and Stata software version 17.0 (StataCorp, College Station, TX, USA). Measurement data that were normally distributed are presented as means ± standard deviations, with group comparisons made using the independent samples T test. For data with skewed distributions, results are expressed using the median (25th percentile, 75th percentile) format, and the Mann–Whitney U test was employed for group comparisons.

The least absolute shrinkage and selection operator (LASSO) regression was used to screen characteristic variables for hypoproteinemia after adult heart valve replacement. Multivariate logistic regression analysis was conducted to identify independent risk factors. The 'st0391_1' package within Stata software facilitated the construction of a nomogram prediction model. The model's discriminatory ability was assessed using the area under the receiver operating characteristic (ROC) curve (AUC), while the Hosmer–Lemeshow test evaluated its calibration. The clinical utility of the nomogram was appraised through decision curve analysis (DCA). To mitigate the risk of overfitting in the nomogram model, bootstrap resampling with 500 iterations was implemented. All statistical tests were two-tailed, with results considered significant at the $P < 0.05$ level.

# RESULTS

## Differential analysis results of hypoproteinemia after adult heart valve replacement

The study population included 143 males and 116 females, with ages ranging from 21 to 85 years and an average age of 58.88 years (±10.97 standard deviation). Based on the incidence of postoperative hypoproteinemia, patients were categorized into two groups: the hypoproteinemia group ($n = 144$) and the non-hypoproteinemia group ($n = 115$). Statistical differences were observed between the two groups in terms of age, history of hypertension BMI, red blood cell (RBC) count, platelet count, and admission albumin levels ($P < 0.05$). The significant association between hypertension and higher incidence of hypoalbuminemia may be attributed to several mechanisms. Hypertension can lead to endothelial dysfunction and increased vascular permeability, potentially contributing to albumin leakage. Additionally, chronic hypertension may exacerbate systemic inflammation or impair renal function, further influencing albumin metabolism. These pathophysiological pathways could explain the observed statistical significance in our study. For further details, refer to Table 1.

## LASSO regression analysis for screening characteristic variables of hypoproteinemia after adult heart valve replacement

Given the large number of research indicators included in this study, to prevent the impact of multicollinearity among different indicators on subsequent multivariate logistic regression and the predictive model, the LASSO regression was employed to reduce the dimensionality of the 27 indicators listed in Table 1. This was done to screen for characteristic variables associated with hypoproteinemia after adult heart valve replacement. A 10-fold cross-validation was performed, and the Lambda.1se value corresponding to

**Table 1  Univariate analysis results of hypoproteinemia after adult heart valve replacement.**

| Variables | Non-hypoproteinemia group | Hypoalbuminemia group | P-value[1] |
|---|---|---|---|
| Number of cases | 115 | 144 | |
| Gender | | | 0.90 |
| Male | 63 (54.8%) | 80 (55.6%) | |
| Female | 52 (45.2%) | 64 (44.4%) | |
| Age (years), mean ± SD | 56.7 ± 11.6 | 60.6 ± 10.1 | 0.004 |
| Education level | | | 0.37 |
| Junior high school and below | 102 (88.7%) | 121 (84.0%) | |
| High school and technical secondary school | 7 (6.1%) | 16 (11.1%) | |
| College or above | 6 (5.2%) | 7 (4.9%) | |
| Retirement | | | 0.25 |
| No | 107 (93.0%) | 128 (88.9%) | |
| Yes | 8 (7.0%) | 16 (11.1%) | |
| Complicated with hypertension | | | <0.001 |
| No | 83 (72.2%) | 73 (50.7%) | |
| Yes | 32 (27.8%) | 71 (49.3%) | |
| Complicated with coronary heart disease | | | 0.41 |
| No | 101 (87.8%) | 131 (91.0%) | |
| Yes | 14 (12.2%) | 13 (9.0%) | |
| Complicated with diabetes | | | 0.064 |
| No | 99 (86.1%) | 134 (93.1%) | |
| Yes | 16 (13.9%) | 10 (6.9%) | |
| Combined with cerebral infarction | | | 0.98 |
| No | 96 (83.5%) | 120 (83.3%) | |
| Yes | 19 (16.5%) | 24 (16.7%) | |
| 2002 NRS nutritional score at admission, median (IQR) | 0.0 (0.0, 1.0) | 0.0 (0.0, 1.0) | 0.85 |
| Body mass index (kg/m2), mean ± SD | 25.7 ± 5.3 | 23.6 ± 2.6 | <0.001 |
| Hemoglobin at admission(g/L), mean ± SD | 129.7 ± 17.5 | 131.9 ± 18.1 | 0.32 |
| White blood cell count at admission ($\times 10^9$/L), mean ± SD | 6.1 ± 2.1 | 5.7 ± 1.6 | 0.089 |
| Red blood cell count at admission ($\times 10^{12}$/L), mean ± SD | 4.5 ± 0.7 | 4.2 ± 0.7 | 0.002 |
| Platelet count at admission ($\times 10^9$/L), mean ± SD | 199.5 ± 67.0 | 169.9 ± 55.2 | <0.001 |
| Serum potassium (mmol/L), mean ± SD | 4.0 ± 0.5 | 4.0 ± 0.4 | 0.22 |
| Serum sodium (mmol/L), mean ± SD | 140.8 ± 2.2 | 140.4 ± 2.4 | 0.15 |
| Serum calcium (mmol/L), median (IQR) | 2.3 (2.2, 2.3) | 2.3 (2.2, 2.3) | 0.90 |
| Serum creatinine (μ, mol/L), median (IQR) | 69.0 (58.0, 80.0) | 71.0 (60.0, 84.5) | 0.11 |
| Serum uric acid (μ, mol/L), median (IQR) | 347.0 (281.4, 403.6) | 351.0 (299.6, 435.3) | 0.26 |
| Albumin at admission(g/L), mean ± SD | 40.0 ± 3.4 | 38.2 ± 4.2 | <0.001 |
| Left ventricular ejection fraction(%), mean ± SD | 62.2 ± 10.0 | 61.9 ± 9.7 | 0.80 |
| Left atrial anteroposterior diameter (mm), mean ± SD | 49.1 ± 10.5 | 48.0 ± 10.5 | 0.40 |
| Left ventricular end systolic diameter (mm), mean ± SD | 38.4 ± 9.4 | 36.9 ± 8.8 | 0.19 |

**Table 1** (*continued*)

| Variables | Non-hypoproteinemia group | Hypoalbuminemia group | *P*-value[1] |
|---|---|---|---|
| Chest infection | | | 0.001 |
| No | 100 (87.0%) | 101 (70.1%) | |
| Yes | 15 (13.0%) | 43 (29.9%) | |
| Pre-operative heart failure | | | <0.001 |
| No | 68 (59.1%) | 52 (36.1%) | |
| Yes | 47 (40.9%) | 92 (63.9%) | |
| Frailty | | | 0.002 |
| No | 72 (62.6%) | 62 (43.1%) | |
| Yes | 43 (37.4%) | 82 (56.9%) | |
| Hemodilution on CPB | | | 0.001 |
| No | 75 (65.2%) | 65 (45.1%) | |
| Yes | 40 (34.8%) | 79 (54.9%) | |

**Notes.**
[1] Independent sample *t* test, Pearson chi-square test, Mann–Whitney rank sum test.

the minimum cross-validation error was selected as the optimal parameter. The indicators with non-zero regression coefficients at this Lambda.1se value were identified as the characteristic variables of hypoproteinemia after adult heart valve replacement.

The LASSO regression results showed that the Lambda.1se value at the minimum cross-validation error was 0.081, and the corresponding characteristic variables included nine indicators: complicated with hypertension, body mass index, red blood cell count at admission, platelet count at admission, albumin level at admission, chest infection, frailty, pre-operative heart failure, and hemodilution on CPB. The detailed results of the LASSO regression analysis are illustrated in Fig. 1.

## Multivariate logistic analysis results of hypoproteinemia after adult heart valve replacement

Utilizing the nine indicators identified by LASSO regression as independent variables, and with the incidence of postoperative hypoproteinemia as the dependent variable, multivariate logistic regression analysis revealed that hypertension, chest infection, frailty and pre-operative heart failure were independent risk factors for the development of hypoproteinemia following heart valve replacement in adults. Conversely, BMI, red blood cell count, platelet count, and admission albumin levels were identified as independent protective factors against postoperative hypoproteinemia. For a comprehensive overview of the associations and statistical values, refer to Table 2.

## Establishment of a nomogram risk prediction model for hypoproteinemia after adult heart valve replacement

In this study, LASSO regression identified nine characteristic variables. Although the *P*-value of the odds ratio (OR) for hemodilution on CPB, which was among the nine variables selected by LASSO regression, in the multivariate logistic regression analysis was >0.05, it was still included in the predictive model. This decision was based on the clinical significance of hemodilution on CPB in relation to hypoproteinemia after adult
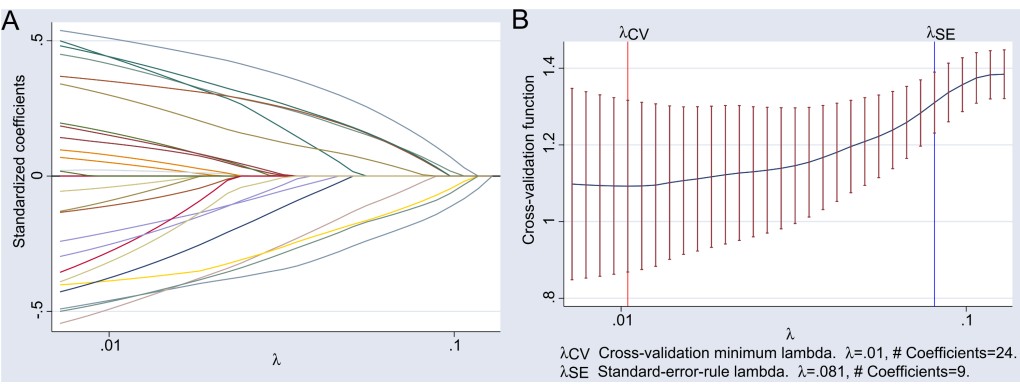

**Figure 1 Results of LASSO regression analysis.** (A) Illustrates the path of standardized coefficients for different variables in LASSO regression as the regularization parameter $\lambda$ varies. The $x$-axis represents the logarithmic scale of the regularization parameter $\lambda$, while the $y$-axis shows the range of standardized coefficient values from $-0.5$ to $0.5$. Each line depicts the trajectory of a variable's coefficient; when $\lambda$ is small (close to 0.01), all 24 coefficients are non-zero. As $\lambda$ increases, some coefficients are shrunk to zero, demonstrating the sparsity characteristic of LASSO regression. (B) Illustrates the cross-validation error as a function of the regularization parameter $\lambda$. The $x$-axis represents the logarithmic scale of the regularization parameter $\lambda$, while the $y$-axis shows the cross-validation function value, indicating the model's performance on the validation set. Two key points are annotated: $\lambda_{CV}$: The value of $\lambda$ corresponding to the minimum cross-validation error, which is $\lambda = 0.01$, associated with 24 non-zero coefficients. $\lambda_{SE}$: The value of $\lambda$ selected based on the standard error rule, which is $\lambda = 0.081$, associated with 9 non-zero coefficients. The shaded area indicates the range of cross-validation error within one standard error, used to assess the stability of model performance.

heart valve replacement, as well as the practical utility of LASSO regression in selecting influential variables for predictive modeling. Following the approach adopted in previous model-building study (*Wenmin, Xiaolong & Yingying, 2020*), all characteristic variables screened by LASSO regression were incorporated into the predictive model, *i.e.,* all nine variables identified by LASSO regression were retained.

Using the partial regression coefficients of these nine indicators derived from the multivariate regression analysis, a nomogram model predicting the risk of hypoproteinemia following adult heart valve replacement was developed using the "nomolog" command from 'st0391_1' package in Stata software (version 17.0). This model is depicted in Fig. 2.

## Clinical applicability analysis of the nomogram model

The DCA curve was utilized to assess the clinical applicability of the nomogram model. The findings indicate that within a threshold probability range of 0.06 to 0.97 for hypoproteinemia in adult patients following heart valve replacement, the nomogram is capable of optimizing the clinical net benefit for patients. This level of benefit significantly exceeds that of both the 'full intervention' and 'no intervention' strategies, thereby demonstrating the nomogram's favorable clinical utility. It is also important to note that the difference in net benefits between the nomogram and the "Treat All" strategy is statistically significant, which is crucial for substantiating the clinical utility of the nomogram. For a more detailed illustration, refer to Fig. 3. Furthermore, compared to the nine predictive indicators listed in Table 2, the nomogram's net benefit curve consistently

**Table 2   Independent sample _t_ test, Pearson chi-square test, Mann–Whitney rank sum test.**

| Variables | β | S.E | Z | _P-value_ | OR (95% CI) |
|---|---|---|---|---|---|
| Complicated with hypertension | | | | | |
| No | | | | | 1.00 (Reference) |
| Yes | 0.85 | 0.32 | 2.62 | 0.009 | 2.34 (1.24∼4.41) |
| Body mass index | −0.13 | 0.05 | −2.94 | 0.003 | 0.87 (0.80∼0.96) |
| Red blood cell count at admission | −0.57 | 0.24 | −2.40 | 0.016 | 0.57 (0.35∼0.90) |
| Platelet count at admission | −0.01 | 0.00 | −2.49 | 0.013 | 0.99 (0.99∼0.99) |
| Albumin level at admission | −0.09 | 0.04 | −2.23 | 0.026 | 0.91 (0.84∼0.99) |
| Chest infection | | | | | |
| No | | | | | 1.00 (Reference) |
| Yes | 1.21 | 0.4 | 3.04 | 0.002 | 3.34 (1.54∼7.28) |
| Frailty | | | | | |
| No | | | | | 1.00 (Reference) |
| Yes | 0.92 | 0.31 | 2.97 | 0.003 | 2.51 (1.37∼4.62) |
| Pre-operative heart failure | | | | | |
| No | | | | | 1.00 (Reference) |
| Yes | 1.28 | 0.32 | 4.02 | <0.001 | 3.60 (1.93∼6.74) |
| Hemodilution On CPB | | | | | |
| No | | | | | 1.00 (Reference) |
| Yes | 0.49 | 0.31 | 1.57 | 0.117 | 1.63 (0.88∼3.02) |

**Notes.**

OR, Odds Ratio; CI, Confidence Interval.

exceeds that of any single indicator across a broader range of threshold probabilities, suggesting that the nomogram has greater clinical applicability than any single indicator.

## Evaluation of discrimination and calibration ability of the nomogram model

The discriminatory ability of the nomogram model was assessed using the area under the receiver operating characteristic (ROC) curve. The result revealed an AUC of 0.823 (95% CI [0.774–0.873]), suggesting that the nomogram possesses good discrimination. For further details, refer to Fig. 4A. In addition, the AUC of the nomogram model is significantly higher than that of any of the nine indicators listed in Table 2. This comparison highlights the added value of the nomogram model over a single-factor model.

The calibration of the nomogram model was evaluated using the Brier score and the Hosmer–Lemeshow goodness-of-fit test. The Brier score for the nomogram was 0.170, which is below the threshold of 0.25, indicating a satisfactory calibration. The Hosmer–Lemeshow test also demonstrated no significant discrepancy between the predicted probabilities and the observed outcomes of postoperative hypoproteinemia (chi square = 5.190, $P = 0.737$), as depicted in Fig. 4B.

To ensure the nomogram's resistance to over-fitting, internal validation was conducted using the Bootstrap resampling method. This internal verification confirmed the nomogram's predictive stability, with a C statistic of 0.796 (95% CI [0.744–0.849]) and a Brier score of 0.248. Additional information can be found in Fig. 4C.

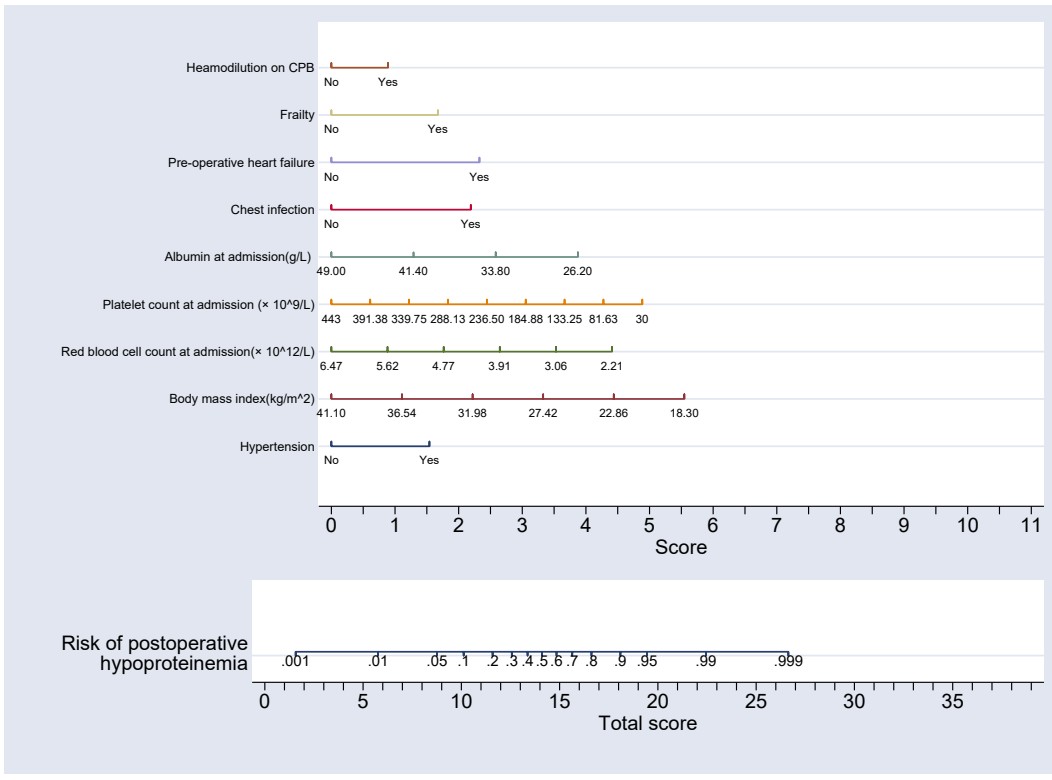

**Figure 2  Nomogram model for the risk of postoperative hypoproteinemia in subjects.** This nomogram integrates various clinical factors to predict the risk of postoperative hypoproteinemia. Variables included in the model are hypertension, body mass index (BMI), red blood cell count at admission, platelet count at admission, albumin level at admission, chest infection, pre-operative heart failure, and hemodilution on cardiopulmonary bypass (CPB). Each variable is assigned a score based on its contribution to the risk, and the total score is calculated by summing the individual scores. The total score is then used to estimate the risk of postoperative hypoproteinemia. Specifically, a vertical line drawn upward from the specific value on the "Total score" axis corresponds to the precise risk value of postoperative hypoproteinemia on the "Risk of postoperative hypoproteinemia" scale, with higher scores indicating a higher risk. This model provides a quantitative tool for clinicians to assess and manage the risk of hypoproteinemia in surgical patients.

## Discussion

Research conducted by Wu Wenjun and colleagues indicates that the duration of endotracheal intubation and hospital stays for patients experiencing hypoproteinemia following cardiopulmonary bypass (CPB) heart surgery are significantly extended, and there is a marked increase in the incidence of postoperative pulmonary infections (*Wenjun et al., 2021*). In the current study, the incidence of hypoproteinemia post-CPB heart valve replacement was 55.60%, which is modestly higher than the rate reported by *Wenjun et al. (2021)*. These findings underscore the importance of focusing on the prevention and management of postoperative hypoproteinemia.

Multivariate logistic regression analysis identified hypertension, chest infection, frailty, and pre-operative heart failure as independent risk factor for the development of hypoproteinemia following heart valve replacement in adults, with an odds ratio (OR)

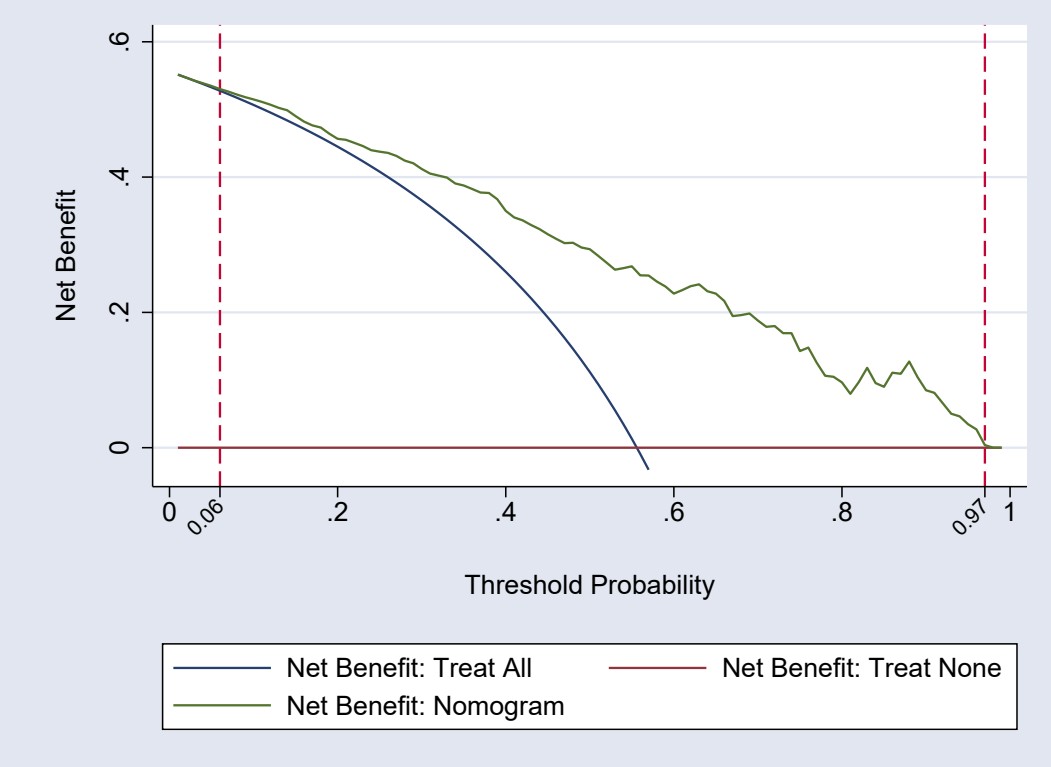

**Figure 3 DCA curve of the nomogram model.** This figure presents the decision curve analysis (DCA) of the nomogram model for predicting postoperative hypoproteinemia. The DCA curve compares the net benefit of using the nomogram model to the strategies of treating all patients or treating none. The *x*-axis represents the threshold probability, which is the probability at which a patient would choose treatment. The *y*-axis shows the net benefit, which is the difference between the true positive rate and the false positive rate, weighted by the threshold probability. The nomogram model demonstrates a higher net benefit compared to the "treat all" and "treat none" strategies across a range of threshold probabilities, indicating its clinical utility in guiding treatment decisions for postoperative hypoproteinemia.

greater than 1 and a statistically significant *P*-value of less than 0.05 (OR > 1, $P < 0.05$). Conversely, BMI, red blood cell count, platelet count, and albumin levels at admission were recognized as independent protective factors against hypoproteinemia, each with an OR less than 1 and a significant *P*-value (OR < 1, $P < 0.05$). Additionally, hemodilution on CPB was also identified by LASSO regression as one of the characteristic variables for the development of hypoproteinemia following heart valve replacement in adults. Hypertension, one of the most prevalent cardiovascular conditions globally, is estimated to affect approximately 16% of China's population (*Minhui & Xiaohong, 2017*). Oxidative stress and vascular endothelial dysfunction have been identified as key contributors to the pathogenesis of hypertension. These factors are intimately linked to the development and progression of the disease (*Popolo et al., 2013*). International cardiologists' research has revealed that the detection rate of micro-albuminuria (MAU) in hypertensive patients is as high as 58.4%. This finding suggests that the permeability of microvessels in hypertensive patients is increased, allowing plasma proteins, which normally do not filter through these vessels, to

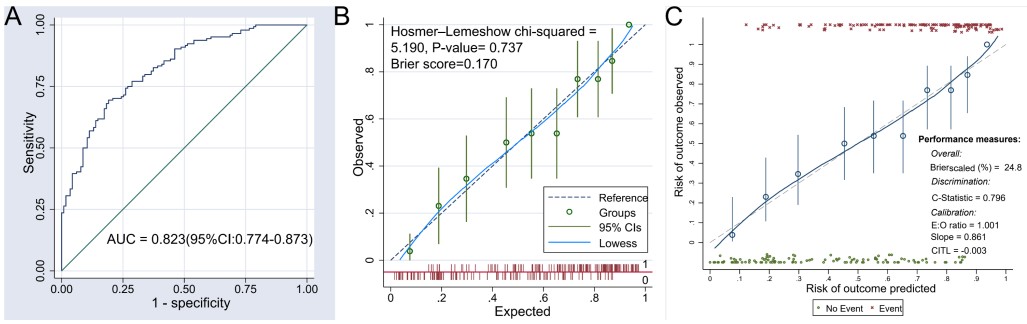

**Figure 4** **ROC curve and calibration curve of the nomogram model.** (A) The receiver operating characteristic (ROC) curve illustrates the diagnostic accuracy of the nomogram model for predicting postoperative hypoproteinemia. The area under the curve (AUC) is 0.823 (95% CI [0.774–0.873]), indicating good discrimination ability. (B) This curve compares the predicted probabilities of postoperative hypoproteinemia with the observed outcomes before internal verification. The Hosmer–Lemeshow chi-square test (Chi-squared = 5.190, $P$-value = 0.737) and Brier score (0.170) suggest good calibration. The Lowess smoothing line (solid) and 95% confidence intervals (dashed) are shown. (C) This curve compares the predicted probabilities with the observed outcomes after internal verification. The calibration metrics include an expected-to-observed (E:O) ratio of 1.001, a slope of 0.861, and a calibration-in-the-large (CITL) value of −0.003. The Lowess smoothing line (solid) and 95% confidence intervals (dashed) are shown. The calibration curves demonstrate the agreement between predicted and observed risks, with the nomogram showing good overall performance (Brier score scaled (%) = 24.8) and discrimination (C-statistic = 0.796).

leak into the interstitial space. Consequently, this leakage leads to a reduction in plasma protein levels (*Böhm et al., 2007*). *Zhenli et al. (2016)* discovered that hypertensive patients exhibit significantly higher levels of Tumor Necrosis Factor-alpha (TNF-α), high-sensitivity C-reactive protein (hs-CRP), and vascular endothelial growth factor (VEGF) compared to patients without hypertension. Moreover, the serum concentrations of TNF-α, hs-CRP, and VEGF in hypertensive patients were found to be positively correlated with average arterial pressure. VEGF, recognized as one of the most potent inflammatory mediators of microvascular permeability, is intricately linked to the development of capillary leakage syndrome. Consequently, this suggests that patients with hypertension experience vascular endothelial injury and an associated increase in vascular permeability. In the context of surgical trauma, particularly with major operations, a substantial release of inflammatory cytokines, such as TNF-α, occurs, creating a 'waterfall effect'. This phenomenon exacerbates the damage to capillary endothelial cells, increasing the likelihood of capillary leakage. Consequently, this can result in hypoproteinemia (*Minhui & Xiaohong, 2017*). In the study by *El Sayed et al. (2014)* it was mentioned that patients developed hypoproteinemia during chest infections due to loss of appetite and malnutrition. In the research conducted by *Zhu et al. (2022)* patients experienced hypoproteinemia as a result of pneumonia and blood infection, with the possible reasons being that the infection exacerbated protein consumption or reduced synthesis. The meta-analysis conducted by *Zhang et al. (2024)* revealed that the risk of frail patients developing hypoproteinemia is 2.37 times higher than that of non-frail patients. This finding is supported by the meta-analysis of *Gangping et al. (2022)* which reported an OR of 1.88, indicating that frailty significantly increases the

risk of postoperative hypoproteinemia. Frail patients often experience a notable decline in physiological reserves and a weakened ability to adapt to stress. Aging leads to reduced muscle mass, diminished organ function, and decreased protein synthesis capacity, while surgical trauma further exacerbates protein catabolism, causing a rapid decline in serum albumin levels (*Zhang et al., 2024*; *Gangping et al., 2022*). Additionally, the frail state is closely associated with nutritional intake disorders. Due to factors such as oral functional decline, limited mobility, or cognitive impairment, patients may face difficulties in eating, loss of appetite, and other issues, resulting in insufficient protein intake. The meta-analysis by *Zhang et al. (2024)* pointed out that frail patients encounter obstacles in food purchasing, preparation, and swallowing, with long-term malnutrition directly increasing the risk of hypoproteinemia. In this study, preoperative heart failure was identified as one of the independent risk factors for postoperative hypoproteinemia. The reason lies in the fact that systemic congestion during heart failure can lead to increased hepatic sinusoidal pressure, impairing the liver's ability to synthesize albumin and resulting in reduced albumin production (*Ruiyu et al., 2017*; *Tan et al., 2024*). Additionally, patients with heart failure often exhibit a chronic inflammatory state (*e.g.*, elevated C-reactive protein), where inflammatory cytokines accelerate protein catabolism while simultaneously suppressing albumin synthesis (*Yixin et al., 2024*). Furthermore, heart failure patients may experience compromised digestive and absorptive functions due to gastrointestinal congestion, leading to insufficient protein intake, and cardiac edema may exacerbate hypoproteinemia (*Jianying, Guanglian & Bangqiang, 2016*).

*Zewei et al. (2025)* discovered that a preoperative platelet count is an independent protective factor against postoperative hypoproteinemia ($OR = 0.995$, $P = 0.028$), a finding that aligns with the outcomes of the current study (*Arques, 2020*). This corroboration further validates the initial observation of this study. Platelets play a multifaceted role, not only in the coagulation process but also in the release of inflammatory mediators and interactions with the vascular endothelium. During surgery or other forms of trauma, increased platelet activation can influence a patient's inflammatory status and protein metabolism. On one hand, elevated platelet levels may signify the body's physiological response to surgical trauma. They contribute to wound healing and tissue repair by releasing growth factors and inflammatory mediators, which could indirectly foster protein synthesis and preserve plasma albumin levels. Furthermore, platelet regulation in inflammatory responses may aid in mitigating postoperative complications, such as infections and tissue damage, which can exacerbate protein breakdown and loss, potentially leading to hypoproteinemia (*Minghua & Hongmei, 2024*). Conversely, platelets have the capacity to protect the vascular endothelium and mitigate the release of inflammatory mediators. This protective function may contribute to the attenuation of postoperative systemic inflammatory response syndrome (SIRS) (*Weifeng, 2020*). SIRS is recognized as a significant precipitant of postoperative hypoproteinemia, as it can escalate protein catabolism, thereby precipitating hypoproteinemia. Related research has identified that a low preoperative albumin level predisposes patients to postoperative hypoproteinemia (*Qiuping, Binru & Xi, 2020*; *Aichao et al., 2024*), a finding that aligns with the observation that the albumin level at admission serves as an independent protective factor against
postoperative hypoproteinemia. Patients presenting with a low preoperative albumin level often exhibit diminished metabolic capacity and suboptimal nutritional status, making them more susceptible to developing postoperative hypoproteinemia. *Junfeng et al. (2024)* reported that BMI with an OR of 0.859 and a *P*-value of 0.021, and preoperative red blood cell count with an OR of 0.424 and a *P*-value of 0.036, are independent protective factors against the development of postoperative hypoproteinemia. These results are in concordance with the findings of the present study. BMI is a widely recognized international standard for assessing obesity and overall health. Patients with a low BMI often exhibit reduced digestive and absorptive capabilities in the gastrointestinal tract, which can diminish the uptake of nutrients essential for albumin synthesis. This, in turn, may lead to a decrease in liver metabolism and consequently, a reduction in albumin production (*Zhang et al., 2017*). Erythrocyte and albumin levels, serving as valid indicators of nutritional outcomes, provide a partial reflection of a patient's nutritional status (*Augustus et al., 2022*). Certain underlying conditions in some patients can precipitate anemia and hypoproteinemia (*Junfeng et al., 2024*).

In the multivariate logistic regression, the OR value for hemodilution on CPB was 1.63 (95% CI [0.88–3.02]). Although the *P*-value for the OR of this indicator was 0.117, hemodilution on CPB was identified by LASSO regression as one of the characteristic variables for the development of hypoproteinemia following heart valve replacement in adults. Hemodilution on CPB leads to a decrease in hematocrit (HCT). If the HCT falls below 22%–24%, it may result in insufficient oxygen delivery, causing ischemia in organs such as the liver, which affects albumin synthesis and contributes to postoperative hypoproteinemia (*Ranucci et al., 2015*). The priming solution (crystalloid or colloid) used in CPB also dilutes the blood, directly reducing plasma albumin concentration (*Wenjun et al., 2021*). Furthermore, the study by *Shufang et al. (2020)* indicated that hemodilution during CPB may trigger a systemic inflammatory response, increasing vascular permeability and causing albumin to leak from the intravascular space into the interstitial tissues, thereby leading to hypoproteinemia.

The nomogram is a graphical statistical tool that integrates multiple predictive variables to estimate the relative risk or probability of a particular outcome. It typically features one or more scales, with each scale representing a distinct variable. By drawing lines to connect the values across these scales, a specific point is determined, which in turn predicts the likelihood of a specific event occurring (*Jing et al., 2023*). The nomogram is extensively utilized in the medical field, aiding healthcare professionals and researchers in assessing disease risks and informing clinical decision-making. Current research indicates that nomograms are capable of predicting the risk of hypoproteinemia in patients suffering from chronic heart failure (*Yixin et al., 2024*). The nomogram has demonstrated effective application in various scenarios, such as estimating the risk of hypoproteinemia in patients following hip joint revision (*Junfeng et al., 2024*) and in cases of postoperative hypoproteinemia in patients with traumatic limb fractures (*Aichao et al., 2024*). However, to date, no research has reported on a risk prediction model specifically for hypoproteinemia in patients post-CPB heart valve surgery, either domestically or internationally. This study addresses this gap by developing a personalized hypoproteinemia risk prediction model based on an analysis

of hypoproteinemia characteristic variables in these patients. The model's discrimination is supported by the area under the ROC curve (AUC = 0.823, 95% CI [0.774–0.873]). Its calibration and predictive accuracy are confirmed by the Hosmer–Lemeshow goodness of fit test and the Brier score, while the DCA curve illustrates the model's clinical applicability. This prediction model is user-friendly and enables real-time prediction of postoperative hypoproteinemia using in-hospital clinical data, facilitating the process for clinical staff to identify high-risk groups, devise intervention strategies, and apply preventive measures.

## LIMITATIONS

There are several limitations to this study that should be acknowledged: (1) As a retrospective cohort study, it is inherently dependent on existing medical records and data, which may be subject to data collection and recording errors, as well as potential incompleteness, thereby risking the introduction of information bias. In this study, pulmonary hypertension, blood transfusions and inotropic support duration were not included in the results analysis due to over 20% missing values during retrospective data collection. These indicators will be included in future prospective studies. (2) The scope of indicators included in this study is constrained, and the analysis does not account for all potential factors influencing hypoproteinemia. The integration of additional relevant indicators could potentially enhance the predictive accuracy of the model, underscoring the importance of broadening the research parameters in future prospective studies. (3) The study's design limits our ability to establish more than a correlation between the indicators and hypoproteinemia; it does not permit the determination of a causal relationship. (4) The constraints of a single-center, retrospective design may affect the statistical power and the generalizability of the prediction model. The study's patient population may not be representative of a wider patient demographic, particularly if the subjects are predominantly from specific regions or healthcare facilities, which could limit the external validity of the findings. Consequently, the nomogram prediction model developed in this study should be subjected to validation within an independent, multi-center, large-sample, prospective cohort study to ascertain its predictive accuracy and practical utility. However, the retrospective nature of this research does not allow for such validation. Specifically, the data collection methods of retrospective studies make it difficult for us to obtain independent, multicenter, prospective datasets with sufficient sample sizes for external validation of the model. (5) Future research could further focus on the development and validation of risk prediction models for chest infection following adult cardiac valve surgery, as numerous additional factors—such as heart failure, pulmonary hypertension, and compromised immunity—are associated with this complication. Conducting such studies would facilitate enhanced postoperative recovery in patients.

## ENLIGHTENMENT TO CLINICAL CARE

Clinically, patients with a low BMI should be subject to heightened vigilance. To mitigate the risk of postoperative hypoproteinemia, it is advisable to strategically increase protein intake preoperatively and ensure timely postoperative nutrition supplementation. To circumvent
hypoproteinemia during the perioperative period, attention should be given not only to albumin levels but also to other nutritional status indicators. For hypertensive patients, close blood pressure monitoring should be implemented pre- and postoperatively, with appropriate management strategies to minimize vascular and tissue damage. Additionally, protein requirements should be assessed, and dietary plans may need adjustment to ensure sufficient protein intake, thereby aiding in the prevention of hypoproteinemia.

In nursing practice, patients with low platelet counts should be closely monitored for hemostatic function, and interventions such as the judicious use of platelet agonists or adjustments to antiplatelet medication dosages should be considered to optimize platelet activity and coagulation status. Furthermore, attention should be paid to the nutritional status of patients, with timely provision of a high-protein diet to support platelet production.

For patients with low red blood cell counts, nurses should regularly evaluate anemia status and implement measures such as iron, vitamin B12, or erythropoietin supplementation to ameliorate anemia. Concurrently, a diet rich in protein and iron should be advocated to facilitate hemoglobin and albumin synthesis.

In cases of low preoperative albumin levels, nursing staff should focus on assessing patients' preoperative nutritional status, particularly for those with low albumin levels. Through preoperative nutritional education, tailored dietary planning, and targeted nutritional supplementation, patients' protein reserves can be enhanced, and the risk of postoperative hypoproteinemia can be diminished. Postoperatively, continuous monitoring of patients' nutritional status is essential, and a personalized nutritional support plan should be formulated.

In addition, during cardiopulmonary bypass, it is essential to strictly control the degree of hemodilution to avoid excessive dilution leading to decreased protein concentration. For patients with preoperative heart failure, proactive optimization of cardiac function and improvement of circulatory status are necessary to reduce postoperative protein loss. Frail patients require enhanced preoperative nutritional assessment and support, along with a high-protein dietary plan to boost protein reserves. Patients with chest infections should receive timely anti-infective treatment to minimize inflammatory consumption, while increasing protein intake to promote recovery. Postoperatively, continuous monitoring of nutritional indicators, combined with personalized nutritional support, is crucial to comprehensively reduce the risk of hypoproteinemia.

The practical utility of the risk prediction model lies in its ability to trigger distinct clinical actions based on specific risk levels. For example, patients with a high predicted risk of postoperative hypoproteinemia can be prioritized for intensive nutritional support and closer monitoring, while those with a lower risk may require only standard care. This approach not only optimizes resource allocation but also enhances the precision of clinical decision-making, bridging the gap between theoretical predictions and actionable interventions.

## CONCLUSION

The rate of postoperative hypoproteinemia among patients with valvular disease at this research center is 55.60%. The nomogram model, formulated by considering the characteristic variables associated with hypoproteinemia, demonstrates adequate calibration and discrimination ability. This model serves as a valuable tool for identifying the high-risk population for hypoproteinemia following valvular surgery. It facilitates the development of targeted intervention strategies designed to mitigate the occurrence of hypoproteinemia post-surgery.

### Funding
The authors received no funding for this work.

### Competing Interests
The authors declare there are no competing interests.

### Author Contributions
- Fang Wang conceived and designed the experiments, analyzed the data, prepared figures and/or tables, and approved the final draft.
- Zhen-zhen Su conceived and designed the experiments, analyzed the data, authored or reviewed drafts of the article, and approved the final draft.
- Xiao-qian Guo conceived and designed the experiments, performed the experiments, prepared figures and/or tables, and approved the final draft.
- Man Li conceived and designed the experiments, performed the experiments, analyzed the data, authored or reviewed drafts of the article, and approved the final draft.
- Rui Wang performed the experiments, analyzed the data, prepared figures and/or tables, and approved the final draft.
- Yan-jun Xu performed the experiments, authored or reviewed drafts of the article, and approved the final draft.

### Human Ethics
The following information was supplied relating to ethical approvals (i.e., approving body and any reference numbers):

The present study obtained approval from the Medical Ethics Committee of the First Affiliated Hospital of the University of Science and Technology of China (Approval number: 2024-RE-210).

### Data Availability
The raw data is available in the Supplementary File.

## Supplemental Information

Supplemental information for this article can be found online at http://dx.doi.org/10.7717/peerj.19676#supplemental-information.

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
