# Peer review of "Development and validation of a risk prediction model for hypoproteinemia after adult cardiac valve surgery: implications for clinical care"

_PeerJ, doi:10.7717/peerj.19676_

## Round 0.1 · original submission · Major Revisions

Please carefully address all the comments of the two reviewers, in particular the concerns from Reviewer 1 regarding the fact that many other factors should be considered

Reviewer 1 ·

Basic reporting

1. The article is well written in clear, unambiguous language with good literature review and references. However some of the references, like Chest infection related to hypoprotenemia need more studies as there are lot more factors associated with chest infection like heart failure, pulmonary hypertension and low immunity than hypoproteinemia alone.
2. The article has shared the necessary tables and structures associated with the statistical analysis.

Experimental design

1. The research question is meaningful but fails to take into account other important factors associated with valve replacement on CPB, like heamodilution on CPB, Pre-operative heart failure, frailty index, which cause hypoproteinemia/hypoalbuminemia.
2. Pre-operative significant factors like Haemoglobin, Platelet count, Serum Albumin can be affected by heart failure which can be seen in many of the valvular heart disease patients especially in patients with pulmonary hypertension and right heart failure.
3. Hence more factors like heart failure, pulmonary hypertension, Frailty Index etc which affect hypoprotenemia should have been considered by the authors and incorporated into the study than the data they considered which can be affected by these factors.

Validity of the findings

1. The statistical analysis of the study is well presented.
2. As mentioned above, many factors which may affect the values considered by the authors as significant predictors of hypoproteinaemia have been missed especially in these patients undergoing double or triple valve replacement who are rheumatic and generally sick and malnourished usually with advanced heart failure. It would be interesting to know, the detailed breakup of the patients who underwent Valve replacement including patients with pulmonary hypertension, pre and postoperative heart failure including inotropic support and blood transfusions during and immediate post-operative periods as all these will affect the haemodilution and thereby the protein levels in the blood and not only the pre-operative factors mentioned by the authors.
3. In conclusion, I believe more data is required for this study to be statistically correct as there are a lot of factors affecting hypopprotenemia post-operatively which have been missed by the authors.

·

Basic reporting

The manuscript is mostly clear, but some statistical explanations could be clearer. Adequate background and literature references are provided, properly situating the study in its field. Structure is standard; however, figure legends are missing and scripts are not shared. The study is self-contained and presents results relevant to the hypotheses but it could be beneficial to include detailed practical applications.

Experimental design

Clearly defined and relevant, addressing a specific knowledge gap in postoperative hypoproteinemia. Conducted rigorously with good ethical standards; Methods are provided but lack complete scripts and details on certain experiments (see details below).

Validity of the findings

Conclusions are clearly linked to the original research question and are well-supported by the results, with appropriate limitations stated to avoid overreaching claims beyond the data shown.

Additional comments

This manuscript introduces a novel predictive model for assessing the risk of postoperative hypoproteinemia in patients undergoing cardiopulmonary bypass heart valve replacement. The primary concern of this review is the figure legends were not provided and the absence of the analytical scripts, which are crucial. Please see the detailed review comments below.

Table 1
(Minor point): The significant association of hypertension with higher incidence of hypoalbuminemia is notable. While this is discussed in the discussion section, integrating a brief explanation of how hypertension could influence postoperative hypoalbuminemia within the results section would provide immediate context for the observed high odds ratio and significant p-value.

Table 2
Using only univariate analysis to select predictors based on statistical significance can overlook intercorrelations among variables and does not establish causality, a point acknowledged in the manuscript's limitations section. However, a more robust feature selection approach would integrate statistical significance with clinical relevance to enhance the model’s practical applicability (understanding the challenges of extracting additional predictive clinical features from EMR systems). Additionally, employing feature selection techniques such as Lasso regression or other machine learning algorithms to determine feature importance could significantly improve the model's capacity to identify truly impactful predictors.

Figure 1 (result section 2.3)
The manuscript lacks detailed interpretation of Figure 1 within the results section.

Figure 2 (result section 2.4 Clinical applicability analysis of the nomogram model)
It would be beneficial for the manuscript to address whether the difference in net benefits between the 'nomogram' and 'Treat All' strategies are statistically significant. This can be crucial for substantiating the clinical utility of the nomogram as recommended by DCA reporting guidelines (Calster et al., 2018).

Figure 3
For Figure 3A (Model Assessment by ROC): An AUC of 0.738 indicates good discriminative ability, but it requires a context or benchmark to fully appreciate its clinical relevance. Although there is no existing model for benchmarking, introducing a simpler model could be effective. For example, using only a single or a few relevant clinical factors, such as hypertension, to predict hypoproteinemia could be valuable. Given that hypertension showed the smartest p-value and odds ratio in the multivariate model and is one of the most prevalent cardiovascular conditions globally (as noted in your discussion section), it would serve well as a baseline model. This baseline model can also be applied to your Figure 2 DCA analysis.

Figure 3C:
While the method section provides the number of bootstrap iterations used, it lacks details on crucial aspects of the bootstrap process. Specifically, it does not clarify whether the model was refitted for each bootstrap sample such as if modifications were made to model parameters during resampling. Additionally, the management of outliers and extreme values during the bootstrap process is not described. Detailed explanations and the script were not provided.
There are plotting errors and scale issues in calibration curve. The manuscript needs to verify that the axes in the calibration plot are correctly scaled from 0 to 1. It appears that some data points (green and red dots) fall outside this typical range, which raises concerns about the validity of these probabilities.

Figure 3A, B and C:
It’s important to do the external validation. Applying the model to an external dataset and comparing the AUC achieved on this new dataset could provide further validation of its robustness and generalizability. The manuscript notes that "the retrospective nature of this research does not allow for such validation." Could the authors clarify the specific barriers, considering its critical role in confirming the model’s applicability?

Introduction and Discussion
The model's ability to predict postoperative hypoproteinemia in real-time is commendable. However, the introduction and discussion sections would benefit from explicitly detailing how these predictions can guide clinical interventions and influence treatment planning. While the "Enlightenment to Clinical Care" section offers valuable guidelines, further illustrating how specific risk levels predicted by the model trigger distinct clinical actions would significantly demonstrate the model’s practical utility and bridge the gap between theoretical predictions and actionable decision-making.

---

## Round 0.2 · accepted · Accept

The authors have adequately addressed all reviewer comments. The revised manuscript has been assessed and is deemed satisfactory and ready for publication.

Reviewer 1 ·

Basic reporting

1. The article is well written in clear, unambiguous language with a good literature review and references.
2. The manuscript now gives a clear definition and background regarding factors affecting poor outcomes
3. The article has shared the necessary tables and structures associated with the statistical analysis.

Experimental design

Though Hypoproteinemia is a known complication following complex cardiac surgeries on cardiopulmonary bypass, to construct and validate a risk prediction model for hypoproteinemia in adults following cardiac valve surgery with cardiopulmonary bypass (CPB) using LASSO regression analysis is a novel concept. The statistical analysis has been detailed and performed to a high technical standard so as to get a good idea about the end points of the research paper.

Validity of the findings

All underlying data is taken into account in the statistical analysis for the regression module, and the prediction of the risk analysis is done robustly. Conclusions are well stated, and the limitations are well explained.

·

Basic reporting

NA

Experimental design

NA

Validity of the findings

NA

Additional comments

The authors have addressed all my concerns and clarified the questions raised in my initial review. I recommend accepting this version of the manuscript.